# Seasonal Change in Microbial Diversity: Bile Microbiota and Antibiotics Resistance in Patients with Bilio-Pancreatic Tumors: A Retrospective Monocentric Study (2010–2020)

**DOI:** 10.3390/antibiotics14030283

**Published:** 2025-03-09

**Authors:** Paola Di Carlo, Nicola Serra, Consolato Maria Sergi, Francesca Toia, Emanuele Battaglia, Teresa Maria Assunta Fasciana, Vito Rodolico, Anna Giammanco, Giuseppe Salamone, Adriana Cordova, Angela Capuano, Giovanni Francesco Spatola, Ginevra Malta, Antonio Cascio

**Affiliations:** 1Department of Health Promotion, Mother and Child Care, Internal Medicine and Medical Specialties, University of Palermo, 90127 Palermo, Italy; paola.dicarlo@unipa.it (P.D.C.); teresa.fasciana@unipa.it (T.M.A.F.); vito.rodolico@unipa.it (V.R.); ginevra.malta@unipa.it (G.M.); antonio.cascio03@unipa.it (A.C.); 2Department of Neuroscience, Reproductive Sciences and Dentistry–Audiology Section, University of Naples Federico II, Via Pansini 5, 80131 Naples, Italy; 3Anatomic Pathology Division, Children’s Hospital of Eastern Ontario, University of Ottawa, Ottawa, ON K1H 8M5, Canada; csergi@cheo.on.ca; 4Department of Laboratory Medicine and Pathology, University of Alberta, Edmonton, AB T6G 2R3, Canada; 5Department of Precision Medicine in Medical, Surgical and Critical Care, University of Palermo, 90127 Palermo, Italy; francesca.toia@unipa.it (F.T.); giuseppe.salamone@unipa.it (G.S.); adriana.cordova@unipa.it (A.C.); 6Endoscopy Unit, University Hospital Paolo Giaccone, 90127 Palermo, Italy; emanuele.battaglia@policlinico.pa.it; 7School of Medicine and Surgery, University of Palermo, 90127 Palermo, Italy; anna.giammanco@unipa.it; 8Department of Emergency, AORN Santobono-Pausilipon, 80122 Naples, Italy; a.capuano@santobonopausilipon.it; 9Department of Biomedicine, Neurosciences and Advanced Diagnostics, 90127 Palermo, Italy; giovannifrancesco.spatola@unipa.it

**Keywords:** bile microbiota, seasonally, Gram-positive, Gram-negative, Candida, antibiotics resistance, bilio-pancreatic tumors

## Abstract

**Background:** Bilio-pancreatic tumors are a severe form of cancer with a high rate of associated mortality. These patients showed the presence of bacteria such as *Escherichia coli* and *Pseudomonas* spp. in the bile-pancreatic tract. Therefore, efficient antibiotic therapy is essential to reduce bacterial resistance and adverse events in cancer patients. Recent studies on the seasonality of infectious diseases may aid in developing effective preventive measures. This study examines the seasonal impact on the bile microbiota composition and the antibiotic resistance of its microorganisms in patients with hepato-pancreatic-biliary cancer. **Methods:** We retrospectively evaluated the effect of the seasonally from 149 strains isolated by 90 Italian patients with a positive culture of bile samples collected through endoscopic retrograde cholangiopancreatography between 2010 and 2020. **Results:** Across all seasons, the most frequently found bacteria were *E. coli*, *Pseudomonas* spp., and *Enterococcus* spp. Regarding antibiotic resistance, bacteria showed the highest resistance to 3GC, fluoroquinolones, aminoglycosides, fosfomycin, and piperacillin-tazobactam in the summer and the lowest resistance in the spring, except for carbapenems and colistin. **Conclusions:** Antibiotic resistance has negative effects in cancer patients who rely on antibiotics to prevent and treat infections. Knowing whether bacterial and fungal resistance changes with the seasons is key information to define adequate and more effective antibiotic therapy.

## 1. Introduction

Cancer is the second leading cause of death in Sicily, with about 12,700 deaths each year. This presents a serious public health issue, primarily for cancers that can be prevented through early diagnosis. In men, the most frequently diagnosed cancers include liver and biliary tract cancers, averaging 487 cases annually, or 3.8% of total cases [1].

The pathogenesis of bilio-pancreatic tumors is complex and involves several key factors. These factors include the abnormal connection between the pancreatic duct and the bile duct, ductal plate malformation in the liver, bile stasis, stone formation, chronic inflammation, and liver fluke infections. Each of these elements can impair the functionality of the bile duct mucosa over time, potentially leading to carcinogenesis. Inflammation is a significant contributor that facilitates the development of epithelial malignancies. Patients may experience complications such as biliary infections, bile duct bleeding, and organ failure [2,3]. Recently, our research team has highlighted the importance of the biliary microbiota, which is distinct from the gut microbiota [3,4,5,6,7].

Understanding the biliary microbiota composition and their resistance patterns is crucial for administering effective antibiotic treatment in patients with biliopancreatic tumors, particularly pancreatic cancer. Tumor growth can block the bile duct, resulting in severe infections. A standard solution is the placement of a biliary stent; however, this intervention can lead to additional complications, especially in patients receiving chemotherapy [8,9].

Seasonal variations are tied to climate factors such as environmental humidity, temperature, atmospheric pressure, and wind, which can impact microbial community composition. These changes create an environment that influences the human microbiota, particularly the microbiota of unsterile organ systems, such as the gut and respiratory microbiota. Seasonal shifts in microbiota can affect its role as a gatekeeper, providing resistance to colonization by pathogens [10,11,12]. Recently, Malta G. et al. [13] published a study on the impact of tuberculosis and seasonality, emphasizing the effects of warming, precipitation, and drought on human infectious diseases. Additionally, there is growing concern about how climate change impacts antimicrobial resistance (AMR) in fungi and bacteria, which become more sensitive to heat exposure due to changing climate conditions, significantly rising ambient temperatures. These microorganisms can be found in natural and hospital environments to opportunistic infections in at-risk groups, including immunocompromised individuals [14,15,16].

There is seasonal variation in AMR rates in community and hospital patients [15,16,17]. This finding may be relevant for our geographical area, where microbiological surveillance data reported a rise in Gram-negative infections attributed to resistant Enterobacteriaceae [18,19,20,21]. This emerging trend presents an opportunity for further research into its potential impact on the bile microbiota of patients with biliopancreatic diseases, especially those diagnosed with cancers. Addressing this issue could lead to improved patient outcomes and better management strategies in these vulnerable populations.

This study aims to evaluate the impact of seasonal variation on the composition of human bile microbiota, particularly in vulnerable populations such as cancer patients. Furthermore, we analyze whether the seasons influence the antibiotic resistance of bacteria found in the biliary samples of our patients.

## 2. Results

The positive bile samples were obtained from 90 consecutive patients with a cancer disease, composed of 55% males and 45% females, with ages 49–94 years old, a mean age of 75.3 years old, and a standard deviation (SD) of 10.1. In Table 1, we reported the general characteristics of the patients enrolled in this study. We generally considered patients with multiple positive bile samples of microorganism isolates.

Table 2 reported the 149 total strain isolates stratified in classes (Gram-negative, Gram-positive, and *Candida* spp.) for seasonally (autumn, summer, winter, and spring). All patients showed one or more strains isolated during bile sampling. Particularly, 54 patients (62%) had a single isolate, and 36 (38%) had more than single isolate. In particular, 54 patients had a single isolate (44 patients had only Gram-negative, 7 patients only Gram-positive, and 3 patients *Candida* spp. only), whereas among 36 with multiple isolates, 16 patients had two isolates (2 patients had 2 Gram-negative, 6 patients had 1 Gram-negative and 1 *Candida* spp., 7 patients had 1 Gram-negative and 1 Gram-positive, and 1 patient had 2 Gram-positive), 18 patients had three isolates (5 patients had 3 Gram-negative, 5 had 2 Gram-negative and 1 Gram-positive, 3 patients had 2 Gram-negative and 1 *Candida* spp., and 5 patients had 1 Gram-negative, 1 Gram-positive, and 1 *Candida* spp.). One patient had four isolates (1 Gram-negative, 2 Gram-positive, and 1 *Candida* spp.), and another one had five isolates (4 Gram-negative and 1 *Candida* spp.).

From Table 2, we found a significant presence of Gram-negative in summer (34.3%, *p* = 0.0197), while a significant presence of *Candida* spp. in spring (52.6%, *p* = 0.0003). Finally, no significant differences among seasons were found about Gram-positive (*p* = 0.86).

Figure 1 and Figure 2 show the individualized negative and positive gram bacteria. Notably, about Gram-negative strains, we found in autumn, the more frequent bacteria were *Escherichia coli* (5.9% = 6/102, *p* = 0.0018) and *Pseudomonas* spp. (8.8% = 9/102, *p* < 0.0001). In summer, the more frequent bacteria were *E. coli* (5.9% = 6/102, *p* = 0.0136), *Klebsiella* spp. (5.9%, = 6/102 *p* = 0.0136), and *Pseudomonas* spp. (7.8% = 8/102, *p* = 0.0001). In winter, the more frequent bacteria were *E. coli* (4.9% = 5/102, *p* = 0.0014) and *Pseudomonas* spp. (6.9% = 7/102, *p* < 0.0001). Finally, in spring, the more frequent bacteria were *E. coli* (3.9% = 4/102, *p* = 0.0014), *Pseudomonas* spp. (2.9% = 3/102, *p* = 0.0465), and *Stenotrophomonas maltophilia* (3.9% = 4/102, *p* = 0.0014).

From Figure 2, in all seasons among Gram-positive bacteria, the most frequent was *Enterococcus* spp. (autumn: 25% = 7/28, *p* < 0.0001; summer: 28.6% = 8/28, *p* < 0.0001; winter: 17.9% = 5/28, *p* < 0.0001; spring: 21.4% = 6/28, *p* < 0.0001).

In Figure 1 and Figure 2, we considered the more frequent bacteria and evaluated the bacteria presence among seasons (autumn, summer, winter, and spring, respectively). Remarkably, no significant differences among seasons were observed for more frequent Gram-negative bacteria such as *E. coli* (5.9%, 5.9%, 4.9%, and 3.9%, *p* = 0.91), *Klebsiella* spp. (2.0%, 5.9%, 2.9%, and 1.0%, *p* = 0.20), *Pseudomonas* spp. (8.8%, 7.8%, 6.9%, and 2.9%, *p* = 0.38), and *S. maltophilia* (2.0%, 2.9%, 2.9%, and 3.9%, *p* = 0.88). Similarly, for more frequent Gram-positive bacteria such as *Enterococcus* spp. (25.0%, 28.6%, 17.9%, and 21.4%, *p* = 0.86).

These results showed that *E. coli* is more common in bile microbiota than others in every season. Still, their predominant presence is constant throughout the seasons.

Table 3 shows the antibiotic resistances, considering the seasonal factors for Gram-negative, Gram-positive, and *Candida* spp., as described in Table 2. Notably, in Table 3, we performed two analyses. The first analysis was reported in the last column, where we compared the specific antibiotic resistance among seasons, and the second analysis, where we investigated which antibiotics showed more presence of bacteria resistance for each season.

In Table 3, we found for all antibiotics, the Gram-negative bacteria identified were more resistant to all antibiotics in summer and less resistant in spring, a part the carbapenem and colistin, where no significant differences in bacteria resistances among seasons were observed. Instead, for Gram-positive bacteria and *Candida* spp., no significant differences among seasons were observed.

From the analysis for each season, among all antibiotics, we found that in autumn the antibiotic with more resistance was the 3GC antibiotic (*p* = 0.0345). In comparison, the antibiotics with less resistance were carbapenem (*p* = 0.031) and colistin (*p* < 0.0001). In summer, we found among antibiotics that colistin was the antibiotic with less resistance compared to others (*p* < 0.0001). In winter, the antibiotic with more resistance was the 3GC antibiotic (*p* = 0.0435), while colistin had less resistance than others (*p* = 0.0002). In spring, no significant differences among antibiotics was observed. Finally, we observed no significant differences among Gram-positive antibiotics for each season.

Additionally, for each season and Gram-negative bacteria, the antibiotics 3GC, cefepime, fluoroquinolones, aminoglycosides, fosfomycin, and piperacillin-tazobactam were more resistant. In comparison, antibiotics such as carbapenem and colistin were the antibiotics with less resistance than others, a part of spring, when the antibiotic with less resistance was only colistin.

In Table 4, we reported the significantly more frequent bacteria individualized from Figure 1 and Figure 2 and considered the antibiotic resistance among seasons. In other words, the Gram-negative bacteria showed greater variability in antibiotic resistance depending on the season.

From Table 4, we observed more frequent bacteria individualized in our study and similar antibiotic resistance among seasons, i.e., no significant differences, while significant differences were observed for each season. Notably, for patients with *E. coli*, we found significantly more resistance to 3GC, cefepime, fluoroquinolones, aminoglycosides, fosfomycin, and piperacillin-tazobactam than colistin in autumn (*p* < 0.05), and for 3GC, cefepime, fluoroquinolones, and piperacillin-tazobactam than colistin in summer (*p* < 0.05).

For *Klebsiella* spp., there was more resistance to 3GC, cefepime, fluoroquinolones, aminoglycosides, fosfomycin, piperacillin-tazobactam, and carbapenem than colistin in autumn (*p* < 0.05), while in winter, more resistance to 3GC, cefepime, fluoroquinolones, and piperacillin-tazobactam than colistin was observed (*p* < 0.05).

For *Pseudomonas* spp., more resistance to 3GC, cefepime, fluoroquinolones, aminoglycosides, fosfomycin, and piperacillin-tazobactam than carbapenem and colistin, and to fosfomycin than colistin in autumn (*p* < 0.05), while in winter, more resistance to 3GC, cefepime, fluoroquinolones, and piperacillin-tazobactam than colistin was observed (*p* < 0.05). In summer and winter, more resistance to 3GC, cefepime, fluoroquinolones, aminoglycosides, fosfomycin, and piperacillin-tazobactam than colistin was observed, while no significant differences were observed in spring.

For *S. maltophilia,* more resistance to 3GC, cefepime, fluoroquinolones, aminoglycosides, fosfomycin, piperacillin-tazobactam, and carbapenem than colistin in summer, winter, and spring was observed.

In conclusion, the antibiotics showed a similar impact across seasons. Particularly in every season, the antibiotics with less resistance were the carbapenem than colistin.

## 3. Discussion

Human gut microbiota, consisting of various microorganisms, is vital for fighting infections and maintaining intestinal balance. It includes bacteria, fungi, viruses, and eukaryotes interacting with the host’s cells and immune system. Disruptions to this microbiota, known as “gut dysbiosis”, can weaken the gut barrier and lead to infections and inflammatory diseases. Recent studies have reported the gut composition in the healthy Italian population [22,23,24].

Gram-negative microorganisms were prevalent in all bile samples of our patients as reported in the literature studies [2,3,4,5,6,7,8,25,26]. Gram-negative bacteria are significantly more resistant to bile acids than Gram-positive bacteria. Primary bile acids are crucial in controlling the overgrowth of harmful Gram-negative bacteria in the small intestine. Bile resistance and tolerance are definitive characteristics that vary among different strains and within individual species [25,26]. Our study found both single and polymicrobial biliary microorganisms. Polymicrobial isolates species were reported during bile sampling through endoscopic retrograde cholangiopancreatography (ERCP), which remains a vital tool for diagnosing and managing both severe disease and/or suspected and confirmed cancer cases such as within our selected cancer population [27,28,29].

In our study, we found a prevalence of Gram-negative bacteria in the summer season. It is known that significantly higher rates of Gram-negative infections were observed during the summer months, compared with other seasons, especially healthcare infections [30,31].

We observed a higher frequency of *Candida* spp. in spring, as reported by other studies that reported the prevalence of *Candida* spp. infections in summertime compared to winter. Changes in temperature, dietary habits, and bodily adaptations due to increased sunlight exposure are potential mechanisms that explain the higher prevalence of *Candida* spp. in skin and mucosal areas during the summer months. This is particularly relevant for elderly and frail individuals such as the patients discussed in our paper. Additionally, the temperature in Southern Italy during spring can disrupt the seasonal wardrobe rotation among the elderly population, creating conditions that favor yeast growth, which can also thrive excessively in warm or humid environments [32,33].

As reported in other studies, we found that the more frequent bacteria responsible for bile dysbiosis in cancer patients were Enterobacteriaceae family, such as the *E. coli*, *Klebsiella* spp., *Pseudomonas* spp., and *Enterococcus* spp. [2,3,4,5,6,7,8,34,35].

The interaction between gut microbiota and bile acids (BAs) is complex and bidirectional. BAs can influence gut microbiota composition, while gut microbiota can alter BA levels. We investigated comorbidities associated with bilio-pancreatic tumors. Most of the concomitant diseases found in our patients were associated with gut dysbiosis, such as liver cirrhosis and diabetes. Gut dysbiosis can compromise the intestinal barrier, increasing inflammation and immune dysfunction, which may raise the risk of cancers like cholangiocarcinoma. When the barrier is disrupted, and bacteria contaminate the bile ducts, toxins can promote cancer development. Bacteria such as *E.* coli in the biliary system and altered bile acid profiles indicate chronic inflammation and elevate the risk of bilio-pancreatic tumors [36,37].

No seasonal differences were found between the most common Gram-negative (*E. coli*, *Klebsiella* spp., and *Pseudomonas* spp.) and Gram-positive (*Enterococcus* spp.) bile isolates.

The analysis of antibiotic resistance and seasonality showed a prevalence of Gram-negative Enterobacteriaceae resistant to most beta-lactams such as third-generation cephalosporins (3GC), cefepime, and piperacillin-tazobactam, and co-resistant to other classes of antibiotics such as fluoroquinolones, aminoglycosides including fosfomycin in the summer season.

These data are following other reports. Studies have explored the notable relationship between elevated outdoor temperatures and the increased incidence of extended-spectrum beta-lactamase (ESBL)-producing Enterobacteriaceae [38,39,40]. This finding suggests a complex interaction that merits further discussion and consideration within the context of public health and environmental factors [40].

Gram-negative bacterial species, such as *P. aeruginosa* and *S. maltophilia*, are primarily sourced from the environment. This suggests a possible connection between environmental temperature and the increasing incidence of antibiotic resistance in these species during summer. These findings may help explain the antibiotic resistance in these bacteria and the isolation of *P. aeruginosa* and *S. maltophilia* in the spring when temperatures rise [41,42,43]. Recent studies involving nursing homes revealed a significant correlation between increasing temperatures and environmental isolation of substantial pathogens such as *K. pneumoniae*. This finding highlights the importance of monitoring temperature changes as a potential factor influencing infection rates, which could inform future public health strategies and intervention efforts [44].

The resistance of Gram-negative bacteria to anti-infective treatments in cancer patients can vary based on factors such as the specific pathogen, geographic region, and healthcare centers. In our study of bile cultures from patients with bilio-pancreatic tumors, we found no significant resistance to carbapenems or colistin, aligning with most studies that report low resistance rates of Gram-negative Enterobacteriaceae to colistin. Cancer patients often receive outpatient care and are generally hospitalized only during significant health declines, frequently undergoing endoscopic retrograde cholangiopancreatography (ERCP) for bile duct obstruction to avoid rapid deterioration and higher mortality rates.

The resistance pattern observed in our data raised concerns. Approximately 30% of the patients we studied underwent biliary stent insertion, which can introduce bacteria during the procedure, potentially leading to infections later when chemotherapy is administered [9].

Our bile data were obtained through cultural investigations aimed at diagnostic and targeted therapy. As previously reported, the culture methodology is preferred for the long-term storage of strains and helps confirm that specific strains responsible for potential infections originate from the microbiota [3].

Our study confirms that resistance to colistin is rare in our geographical area. Colistin is presently considered the last line of defense against human infections caused by multidrug-resistant Gram-negative organisms, and in the old era, colistin therapy was at least as effective and as safe a beta-lactam antibiotic or a quinolone in the treatment of infections caused by multidrug-resistant such as *P. aeruginosa* and useful or preferred alternative therapy for this infection in cancer patient. Nowadays, the availability of newer agents and combination therapy for Gram-negative ESBL and metallo-beta-lactamase (MBL) produce Enterobacterales and its potential nephrotoxicity and neurotoxicity but make colistin treatment less desirable [45,46,47]. In light of the growing concerns surrounding colistin resistance, it is important to prioritize continuous surveillance and implement preventive strategies. Such measures play a crucial role in safeguarding public health and ensuring the effectiveness of our therapeutic options. Italy is engaged in European surveillance networks and local initiatives like the AR-ISS network, improving its capacity to detect AMR trends in real-time. The country aims to establish a sustainable, multi-sectoral strategy addressing AMR that aligns with the “One Health” approach, considering human, animal, and environmental health [48].

### Limitations

The advanced age at hospitalization (mean: 75.5 years), the short hospitalization period (mean: 15.6 days), and the high mortality rate of about 53% at 6 months, as reported by Di Carlo P. et al. [4] in the Sicilian region, make it difficult to follow the same patients across seasons. Despite this gap, the authors believe that the results obtained are essential to provide a detailed situation of the impact of seasonality on isolate species and antibiotic resistance to improve preventive measures such as antimicrobial stewardship and hospital control [48,49,50].

Regarding the use of ERCP (endoscopic retrograde cholangiopancreatography), it is an important procedure used to address blockages in the bile or pancreatic ducts and to obtain cell or tissue samples. While it can be highly effective, it is essential to be aware of potential complications, such as infection, bleeding, perforation, or pancreatitis (inflammation of the pancreas). Due to these risks, collecting bile samples from the same patient at different times to study seasonal variations in bile microbiota composition presents challenges. This highlights the need for alternative approaches and careful planning in future research to better understand these variations without compromising patient safety [51,52,53].

This is a retrospective study conducted at a single center. Additionally, the authors observed in the statistical analyses the impact due to a small sample, and some no significant results were obtained.

Therefore, the authors are planning to perform a prospective ESCMID multicenter study, involving several countries. Nevertheless, this study is going to focus on some remarkable data, which are associated with seasonality.

## 4. Materials and Methods

We retrospectively evaluated the impact of the seasons on bile microbiota in bile samples from 90 with bilio-pancreatic tumors, including cholangiocarcinoma, gallbladder carcinoma, and pancreatic cancer enrolled at the Department of General Surgery of Policlinic University Hospital “Paolo Giaccone (A.U.O.P.), University of Palermo, Italy, from 1 January 2010 to 31 December 2020. The study population consisted of Italian patients referred to endoscopic retrograde cholangiopancreatography (ERCP) for treatment of bile duct obstruction according to standardized hospital recommendations [54].

Patients with laboratory or clinical signs of sepsis using the definitions of the Third International Consensus for Sepsis and Septic Shock (Sepsis-3) were excluded from the study [55].

Inclusion criteria: performance of ERCP with at least one positive bile sample for fungal or bacteria and/or cyto- and/or histopathological examination. In the period of patients’ recruitment, the number of endoscopic procedures carried out was 250/year on average (50% from individuals from other hospitals of the Sicilian region). The bile collection was obtained by introducing an ERCP standard 5 Fr catheter (Olympus Medical Systems Co. Tokyo, Japan) deeply into the common bile duct (CBD), before the injection of the contrast medium. In the case of sepsis due to plastic stent obstruction, the biliary sample was obtained immediately after the endoscopic stent removal.

All biliary and pancreatic system diseases were extracted from medical charts and double-checked by two team specialists.

All enrolled patients for treatment of bile duct obstruction were diagnosis-naïve (n = 9), subjects with early cancer diagnosis in chemotherapy (n = 16/90) and without chemotherapy (28/90), and subjects who underwent ERCP procedure for preoperative biliary decompression (9/10), palliative endoscopic stents (28/90).

Medical comorbidities such as hypertension, presence of diabetes, chronic heart failure (CHF), chronic obstructive pulmonary disease (COPD), dementia (DM), liver cirrhosis (LC), hemodialysis (HD), chronic renal failure (CRF), autoimmune diseases (ADs) were identified according to Italian ICD-9-CM codes.

Only 2/90 (2.22%) of enrolled patients showed infection after ERCP due to Gram-negative MDR pathogens (carbapenem-resistant Klebsiella pneumoniae) [19,20,21].

A surveillance program for multidrug-resistant Gram-negative bacilli, including active surveillance cultures, has been carried out in the surgical emergency unit since January 2010 [18,19]. All patients are routinely screened on admission, and a model for implementing targeted infection control strategies and antibiotic stewardship is active in A.U.O.P’s policy.

Bacterial identification and antimicrobial susceptibility testing were carried out using either the Phoenix Automated Microbiology System (Becton Dickinson Diagnostic Systems, Sparks, NV, USA) or the Vitek-2 System (Bio-Mérieux, Marcy l’Etoile, France). The antimicrobial sensitivity test of the strains was determined according to the European Committee on Antimicrobial Susceptibility Testing (EUCAST) breakpoints as previously reported [3,4,5,6,7]. As previously reported, *Candida* spp. was also identified by conventional morphological and biochemical methods [3,5].

All procedures performed in studies involving human participants were in accordance with the ethical standards of the institutional and/or national research committee and with the 1964 Helsinki Declaration and its later amendments or comparable ethical standards. The Regional Ethics Board approved this study (REB: AIFA code–CE 150109; protocol N° 09/2021).

### Statistical Analysis

Data are presented as numbers and percentages for categorical variables and continuous data are expressed as mean ± standard deviation (SD), and median and interquartile interval (IRQ = [Q1; Q3]).

Multiple comparison Chi-square test was used to define significant differences among three or more independent variables. If the Chi-square test was significant (*p* < 0.05), a post hoc Z-test was performed to identify the highest or lowest significant frequency. Fisher’s exact test was used, but the Chi-square test was not appropriate. The Chi-square goodness of fit was used to evaluate significant differences among three or more variable modalities.

Cochran’s Q test was used to test the differences between three or more paired frequencies or proportions. When the Cochran’s Q test was positive (*p* < 0.05), a minimum required difference for a significant difference between two proportions was calculated according to the Bonferroni–Dunn method. All data were analyzed using the MATLAB statistical toolbox version 2008 (MathWorks, Natick, MA, USA) for 32-bit Windows.

## 5. Conclusions

New research indicates that more attention should be focused on how intestinal pathogens undermine protective mechanisms, highlighting the increasing understanding of how metabolic factors can act as critical virulence determinants that overcome colonization resistance [56,57,58]. Factors such as lifestyle, including dietary factors, medications, and co-morbidities continually alter microbiota composition. Studying these changes can enhance personalized medicine, particularly in frailty subjects, though further research is needed to clarify the microbiota–disease relationship. Antibiotics can disrupt microbiota, increasing the risk of infections and antimicrobial resistance, emphasizing the need to better understand these connections, especially for immuno-compromised patients like those with cancer [8,38,59].

Understanding the seasonal incidence patterns of bacterial pathogens can enhance surveillance of endemic diseases, improve preparedness for outbreaks, and guide public health interventions. Most studies on the seasonal patterns of bacterial pathogens have been conducted in acute care settings, with less attention given to community-based data. Additionally, data from high-risk patient populations, such as those with hepato-biliary and pancreatic disorders, including cancer, is rarely included in these studies.

The impact of seasonality on environmental contamination is a crucial factor, as it indicates patient colonization and dysbiosis in vulnerable populations. Therefore, investigating these factors’ seasonal patterns and the burden on patients is essential.

Using longitudinal microbial surveillance data from patients and their immediate environments, we aimed to examine the seasonal patterns of both patient and environmental burdens caused by several common bacterial pathogens in nursing homes (NHs). We also compared the seasonal variations between antibiotic-resistant and susceptible isolates and assessed their associations.

## Figures and Tables

**Figure 1 antibiotics-14-00283-f001:**
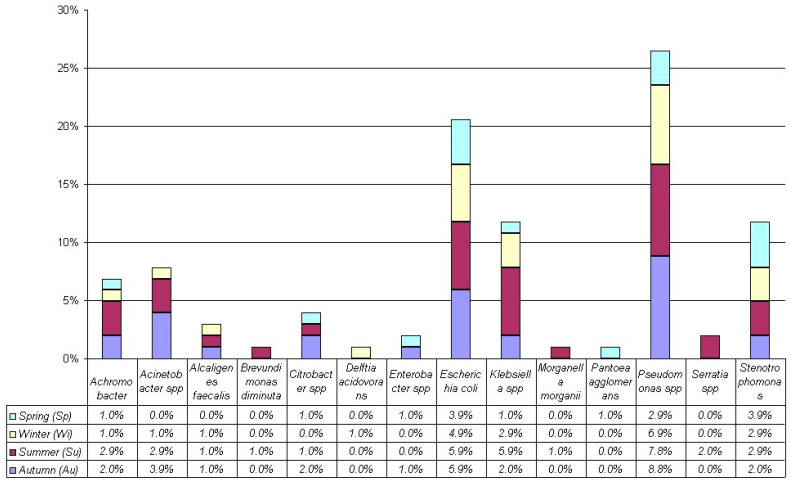
Gram-negative isolated from 90 patients.

**Figure 2 antibiotics-14-00283-f002:**
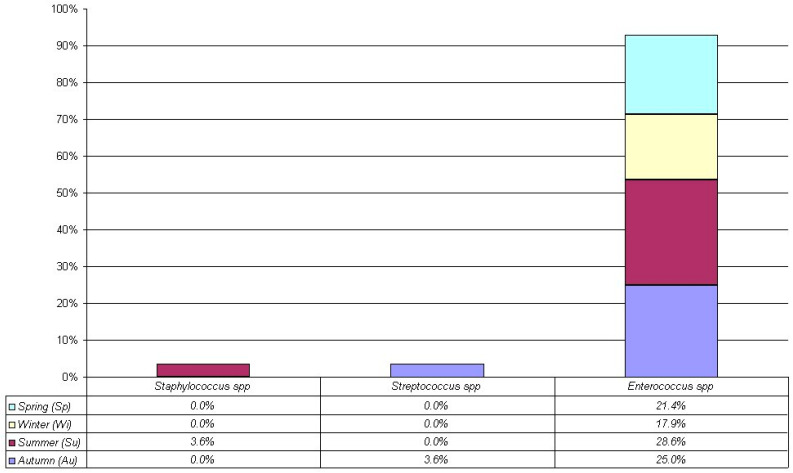
Gram-positive isolated from 90 patients.

**Table 1 antibiotics-14-00283-t001:** General characteristics of 90 patients, including bile isolates.

Parameters	Sample
*Patients*	90
*Age at hospitalization*	
Mean ± SD	75.5 ± 10.1
Median (IQR)	76.0 (70.0, 83.0)
*Time of hospitalization* (*days*)	
Mean ± SD	15.6 ± 22.2
Median (IQR)	8.0 (4.0, 17.0)
*Gender*	
Male	58.9% (53)
Female	41.1% (37)
*Isolates **	
*Gram* −	87.8% (79)
*Gram* +	31.1% (28)
*Candida* spp.	21.1% (19)
*Tumor type*	
Gallbladder carcinoma	5.6% (5)
Cholangiocarcinoma	28.9% (26)
Pancreatic cancer	65.6% (59)
*Comorbidity* †	
Acute coronary syndrome (ACS)	20.0% (18)
Autoimmune diseases (AD)	8.9% (8)
Chronic heart failure (CHF)	34.4% (31)
Chronic obstructive pulmonary disease (COPD)	15.6% (14)
Chronic renal failure (CRF)	20.0% (18)
Dementia (DM)	15.6% (14)
Diabetes	48.9% (44)
Hypertension	57.8% (52)
Hemodialysis (HD)	3.3% (3)
Liver cirrhosis (LC)	32.2% (29)

SD = standard deviation; IQR = interquartile range; Gram − = Gram-negative; gram + = Gram-positive; * = many patients had more bacteria, both Gram +, Gram −, and *Candida* spp.; † = more patients had more than one comorbidity.

**Table 2 antibiotics-14-00283-t002:** Strains isolated stratified for groups from 90 patients. In particular, in the last column, we report the statistical analyses among seasons for each isolate.

Isolated Strains	Total	Autumn (Au)	Summer (Su)	Winter (Wi)	Spring (Sp)	Among Seasons *p*-Value
Total	149	27.5% (41/149)	30.9% (46/149)	20.8% (31/149)	20.8% (31/149)	
*Gram −*	68.5% (102/149)	28.4% (**29/102**)	34.3% (35/102)	21.6% (22/102)	15.7% (16/102)	*p* = 0.0353 * (Cgf) Summer **, *p* = 0.0197 (Z)
*Gram +*	18.8% (28/149)	28.6% (8/28)	32.1% (9/28)	17.9% (5/28)	21.4% (6/28)	*p* = 0.86 (Cgf)
*Candida* spp.	12.8% (19/149)	21.1% (4/19)	10.5% (2/19)	15.8% (3/19)	52.6% (10/19)	*p* = 0.0429 ***** (Cgf) Spring **, *p* = 0.0003 (Z)

* = significant test; ** = modality more frequent; Cgf = Chi-square goodness of fit; Z = post hoc Z-test was performed if Chi-square goodness fit was significant.

**Table 3 antibiotics-14-00283-t003:** Antibiotic resistances considering the seasonally for the isolate categories.

Isolates/Antibiotics	TOTAL Resistance	Autumn (Au) % (n)	Summer (Su) % (n)	Winter (Wi) % (n)	Spring (Sp) % (n)	Among Seasons *p*-Value (Test)
*Gram −* (102)	n = 102	28.4% (29/102)	34.3% (35/102)	21.6% (22/102)	15.7% (16/102)	
(1) 3GC	94.1% (96)	100% (29)	94.3% (33)	90.9% (20)	87.5% (14)	*p* = 0.0261 * (Cgf) Su **, *p* = 0.0226 Sp ***, *p* = 0.0264
(2) Cefepime	82.4% (84)	93.1% (27)	91.4% (32)	72.7% (16)	56.3% (9)	*p* = 0.0014 * (Cgf) Su **, *p* = 0.0021
(3) Fluoroquinolones	88.2% (90)	96.6% (28)	94.3% (33)	86.4% (19)	62.5% (10)	*p* = 0.0033 * (Cgf) Su **, *p* = 0.0052 Sp ***, *p* = 0.0053
(4) Aminoglycosides	71.6% (73)	79.3% (23)	74.3% (26)	68.2% (15)	56.3% (9)	*p* = 0.0204 * (Cgf) Su **, *p* = 0.0228 Sp ***, *p* = 0.02
(5) Fosfomycin	69.6% (71)	75.9% (22)	77.1% (27)	63.6% (14)	50.0% (8)	*p* = 0.0074 * (Cgf) Su **, *p* = 0.0050 Sp ***, *p* = 0.0134
(6) Piperacillin- tazobactam	78.4% (80)	79.3% (23)	91.4% (32)	72.7% (16)	56.3% (9)	*p* = 0.0023 * (Cgf) Su **, *p* = 0.0005 Sp ***, *p* = 0.0086
(7) Carbapenem	40.2% (41)	37.9% (11)	45.7% (16)	31.8% (7)	43.8% (7)	*p* = 0.15 (Cgf)
(8) Colistin	0.0% (0)	0.0% (0)	0.0% (0)	0.0% (0)	0.0% (0)	*p* = N/A
**Analysis in the season** ***p*-value (test)**	*p* < 0.001 * (Q) (1), (2), (3), (4), (5), (6) > (7) **, *p* < 0.05 (B) (1), (2), (3), (4), (5), (6) > (8) **, *p* < 0.05 (B)	*p* < 0.001 * (Q) (1), (2), (3), (4) and (6) > (7) **, *p* < 0.05 (B) (1), (2), (3), (4), (5), (6), (7) > (8)**, *p* < 0.05 (B)	*p* < 0.001 * (Q) (1), (2), (3) and (6) > (7) **, *p* < 0.05 (B) (1), (2), (3), (4), (5), (6), (7) > (8) **, *p* < 0.05 (B)	*p* < 0.001 * (Q) (1) > (7) **, *p* < 0.05 (B) (1), (2), (3), (4), (5), (6) > (8) **, *p* < 0.05 (B)	*p* < 0.001 * (Q) (1), (2), (3), (4), (5), (6) > (8) **, *p* < 0.05 (B)	
*Gram +* (28)	n = 28	28.6% (8/28)	32.1% (9/28)	17.9% (5/28)	21.4% (6/28)	
(1) Oxacillin	10.7% (3)	25.0% (2)	0.0% (0)	0.0% (0)	16.7% (1)	*p* = N/A
(2) Ampicillin	39.3% (11)	50.0% (4)	22.2% (2)	40.0% (2)	50.0% (3)	*p* = 0.80 (Cgf)
(3) Vancomycin	28.6% (8)	25.0% (2)	33.3% (3)	40.0% (2)	16.7% (1)	*p* = N/A
(4) Gentamicin	32.1% (9)	37.5% (3)	22.2% (2)	20.0% (1)	50.0% (3)	*p* = N/A
**Analysis in** **the season** ***p*-value (test)**	*p* = 0.19 (Q)	N/A	N/A	N/A	N/A	
*Candida* spp. (19)	n = 19	21.1% (4/19)	10.5% (2/19)	15.8% (3/19)	2.6% (10/19)	
(1) Fluconazole	10.5% (2)	50.0% (2)	0.0% (0)	0.0% (0)	0.0% (0)	*p* = N/A
(2) Echinocandins	0.0% (0)	0.0% (0)	0.0% (0)	0.0% (0)	0.0% (0)	*p* = N/A
**Analysis in** **the season** ***p*-value (test)**	N/A	N/A	N/A	N/A	N/A	

3GC = third-generation cephalosporins; * = significant test; ** = modality more frequent; *** = modality less frequent; Cgf = Chi-square goodness of fit; Q = Cochran’s Q test; B = post hoc test by Bonferroni–Dunn method for pairwise comparison; N/A = there were insufficient data to perform the test.

**Table 4 antibiotics-14-00283-t004:** Antibiotic resistances considering the seasonal variation for more frequent bacteria of Table 2.

Isolates Bacteria	TOTAL Resistance	Autumn (Au) % (n)	Summer (Su) % (n)	Winter (Wi) % (n)	Spring (Sp) % (n)	Among Seasons *p*-Value (Test)
*E. coli* (n = 21)		n = 6	n = 6	n = 5	n = 4	
(1) 3GC	90.5% (19)	100% (6)	100% (6)	80.0% (4)	75.0% (3)	*p* = 0.70 (Cgf)
(2) Cefepime	81.0% (17)	100% (6)	100% (6)	60.0% (3)	50.0% (2)	*p* = 0.39 (Cgf)
(3) Fluoroquinolones	85.7% (18)	100% (6)	100% (6)	80.0% (4)	50.0% (2)	*p* = 0.49 (Cgf)
(4) Aminoglycosides	61.9% (13)	83.3% (5)	50.0% (3)	60.0% (3)	50.0% (2)	*p* = 0.69 (Cgf)
(5) Fosfomycin	57.1% (12)	83.3% (5)	50.0% (3)	40.0% (2)	50.0% (2)	*p* = 0.57 (Cgf)
(6) Piperacillin-tazobactam	81.0% (17)	100% (6)	100% (6)	60.0% (3)	50.0% (2)	*p* = 0.39 (Cgf)
(7) Carbapenem	19.0% (4)	33.3% (2)	16.7% (1)	0.0% (0)	25.0% (1)	*p* = N/A
(8) Colistin	0.0% (0)	0.0% (0)	0.0% (0)	0.0% (0)	0.0% (0)	*p* = N/A
**Analysis in** **the season** ***p*-value (test)**	*p* < 0.001 * (Q) (1), (2), (3), (6) > (7), (8), *p* < 0.05 * (B) (4), (5) > (8), *p* < 0.05 * (B)	*p* < 0.001 * (Q) (1), (2), (3), (4), (5), (6) > (8), *p* < 0.05 * (B)	*p* < 0.001 * (Q) (1), (2), (3), (6) > (8), *p* < 0.05 * (B)	N/E	N/E	
*Klebsiella* spp. (n = 12)		n = 2	n = 6	n = 3	n = 1	
(1) 3GC	91.7% (11)	100% (2)	83.3% (5)	100% (3)	100% (1)	*p* = 0.36 (Cgf)
(2) Cefepime	83.3% (10)	100% (2)	83.3% (5)	66.7% (2)	100% (1)	*p* = 0.31 (Cgf)
(3) Fluoroquinolones	91.7% (11)	100% (2)	83.3% (5)	100% (3)	100% (1)	*p* = 0.36 (Cgf)
(4) Aminoglycosides	66.7% (8)	50.0% (1)	66.7% (4)	66.7% (2)	100% (1)	*p* = N/A
(5) Fosfomycin	66.7% (8)	50.0% (1)	66.7% (4)	66.7% (2)	100% (1)	*p* = N/A
(6) Piperacillin-tazobactam	75.0% (9)	50.0% (1)	83.3% (5)	66.7% (2)	100% (1)	*p* = N/A
(7) Carbapenem	58.3% (7)	50.0% (1)	50.0% (3)	66.7% (2)	100% (1)	*p* = N/A
(8) Colistin	0.0% (0)	0.0% (0)	0.0% (0)	0.0% (0)	0.0% (0)	*p* = 1.0
**Analysis in** **the season** ***p*-value (test)**	*p* < 0.001 * (Q) (1), (2), (3), (4), 5), (6), (7) > (8), *p* < 0.05 * (B)	*p* = 0.18 (Q)	*p* < 0.001 * (Q) (1), (2), (3), (6) > (8), *p* < 0.05 * (B)	*p* = 0.051 (Q)	N/E	
*Pseudomonas* spp. (n = 27)		n = 9	n = 8	n = 7	n = 3	
(1) 3GC	96.4% (26)	100% (9)	100% (8)	85.7% (6)	100% (3)	*p* = 0.36 (Cgf)
(2) Cefepime	85.2% (23)	100% (9)	100% (8)	71.4% (5)	33.3% (1)	*p* = 0.081 (Cgf)
(3) Fluoroquinolones	85.2% (23)	100% (9)	100% (8)	71.4% (5)	33.3% (1)	*p* = 0.081 (Cgf)
(4) Aminoglycosides	81.5% (22)	88.9% (8)	100% (8)	71.4% (5)	33.3% (1)	*p* = 0.11 (Cgf)
(5) Fosfomycin	81.5% (22)	88.9% (8)	100% (8)	71.4% (5)	33.3% (1)	*p* = 0.11 (Cgf)
(6) Piperacillin-tazobactam	77.8% (21)	77.8% (7)	100% (8)	71.4% (5)	33.3% (1)	*p* = 0.14 (Cgf)
(7) Carbapenem	33.3% (9)	22.2% (2)	50.0% (4)	28.6% (2)	33.3% (1)	*p* = N/A
(8) Colistin	0.0% (0)	0.0% (0)	0.0% (0)	0.0% (0)	0.0% (0)	*p* = N/A
**Analysis in** **the season** ***p*-value (test)**	*p* < 0.001 * (Q) (1), (2), (3), (4), (5), (6) > (7), (8), *p* < 0.05 * (B) (7) > (8), *p* < 0.05 * (B)	*p* < 0.001 * (Q) (1), (2), (3), (4), (6) > (7), (8), *p* < 0.05 * (B) (5) > (8), *p* < 0.05 * (B)	*p* < 0.001 * (Q) (1), (2), (3), (4), (5), (6) > (8), *p* < 0.05 * (B)	*p* = 0.002 * (Q) (1), (2), (3), (4), (5), (6) > (8), *p* < 0.05 * (B)	*p* = 0.072 (Q)	
*S. maltophilia* (n = 12)		n = 2	n = 3	n = 3	n = 4	
(1) 3GC	100% (12)	100% (2)	100% (3)	100% (3)	100% (4)	*p* = 0.88 (Cgf)
(2) Cefepime	100% (12)	100% (2)	100% (3)	100% (3)	100% (4)	*p* = 0.88 (Cgf)
(3) Fluoroquinolones	100% (12)	100% (2)	100% (3)	100% (3)	100% (4)	*p* = 0.88 (Cgf)
(4) Aminoglycosides	100% (12)	100% (2)	100% (3)	100% (3)	100% (4)	*p* = 0.88 (Cgf)
(5) Fosfomycin	100% (12)	100% (2)	100% (3)	100% (3)	100% (4)	*p* = 0.88 (Cgf)
(6) Piperacil-lin-tazobactam	100% (12)	100% (2)	100% (3)	100% (3)	100% (4)	*p* = 0.88 (Cgf)
(7) Carbapenem	75.0% (9)	50.0% (1)	100% (3)	66.7% (2)	75.0% (3)	*p* = N/A
(8) Colistin	0.0% (0)	0.0% (0)	0.0% (0)	0.0% (0)	0.0% (0)	*p* = N/A
**Analysis in** **the season** ***p*-value (test)**	*p* < 0.001 * (Q) (1), (2), (3), (4), (5), (6), (7) > (8), *p* < 0.05 * (B)	*p* = 0.12 (Q)	*p* = 0.004 * (Q) (1), (2), (3), (4), (5), (6), (7) > (8), *p* < 0.05 * (B)	*p* = 0.016 * (Q) (1), (2), (3), (4), (5), (6), (7) > (8), *p* < 0.05 * (B)	*p* = 0.016 * (Q) (1), (2), (3), (4), (5), (6), (7) > (8), *p* < 0.05 * (B)	
*Enterococcus* spp. (n = 24)		n = 7	n = 7	n = 4	n = 6	
(1) Oxacillin	8.3% (2)	14.3% (1)	0.0% (0)	0.0% (0)	16.7% (1)	*p* = N/A
(2) Ampicillin	45.8% (11)	57.1% (4)	28.6% (2)	50.0% (2)	50.0% (3)	*p* = 0.80
(3) Vancomycin	33.3% (8)	28.6% (2)	42.9% (3)	50.0% (2)	16.7% (1)	*p* = N/A
(4) Gentamicin	37.5% (9)	42.9% (3)	28.6% (2)	25.0% (1)	50.0% (3)	*p* = N/A
**Analysis in** **the season** ***p*-value (test)**	*p* < 0.001 * (Q) (2) > (1), *p* < 0.05 * (B)	*p* = 0.34 (Q)	*p* = 0.10 (Q)	*p* = 0.19 (Q)	*p* = 0.45 (Q)	

* = significant test; Cgf = Chi-square goodness of fit; Q = Cochran’s Q test; B = post hoc test by Bonferroni–Dunn method for pairwise comparison; N/A = there were insufficient data to perform the test.

## Data Availability

Dataset available on request from the project Manager, Prof. Paola Di Carlo.

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
