# Peer review of "Seasonal Change in Microbial Diversity: Bile Microbiota and Antibiotics Resistance in Patients with Bilio-Pancreatic Tumors: A Retrospective Monocentric Study (2010–2020)"

_antibiotics, 2025, doi:10.3390/antibiotics14030283_

Round 1
Reviewer 1 Report
Comments and Suggestions for Authors
This is a very interesting and informative retrospective study regarding seasonal impact on the bile microbiota composition and the antibiotic resistance of its microorganisms in patients with hepato-pancreatic-biliary cancer. To increase to interest for the readers, I suggest following revisions of the manuscript.
The authors mention that their investigation was done in patients with hepato-pancreatic-biliary cancer. However, patients with liver cancer are not included in the study.
It is not clear what time of hospitalization covers.
It is not clear of the patients are treatment naïve or they were included in the study after different treatments for cancer, such chemotherapy. Also, a discussion regarding the impact of chemotherapy may have on shaping bile microbiota and developing resistance to antibiotics.
It is also not clear what was the incidence of infections from hepato-pancreatic-biliary anatomic regionin the patients before enrolling in the study.
The authors should include in their demographic table co-morbidities of patients and their corresponding treatments along with a discussion how these two parameters may influence the gut microbiota and resistance to antibiotics.
Since gut microbiota the bile microbiota can impact each other, the study would have benefited. However, a strong discussion related to this topic should be added at least.
Importantly, a discussion about the composition of gut microbiota in healthy people from the same geographic space should be added to help reader to understand the changes in gut microbiota in the study patients.
Finally, there are a lot of tinny editing issues and the manuscript needs a professional proof of English proficiency.
Comments on the Quality of English LanguageThe manuscript needs a professional proof of English proficiency.
Author Response
REVIEWER 1
This is a very interesting and informative retrospective study regarding seasonal impact on the bile microbiota composition and the antibiotic resistance of its microorganisms in patients with hepato-pancreatic-biliary cancer. To increase to interest for the readers, I suggest following revisions of the manuscript.
1) The authors mention that their investigation was done in patients with hepato-pancreatic-biliary cancer. However, patients with liver cancer are not included in the study.
[Reply]: Thank you for your question. In the study, we included Cholangiocarcinoma, a rare malignant tumour of the liver, which originates from the cells that make up the bile ducts. However, the authors changed the term “hepato-bilio-pancreatic” cancer to “bilio-pancreatic tumors”, in the title and main text.
2) It is not clear what time of hospitalization covers.
[Reply]: Thank you for your suggestion. We added in Methods section.
3) It is not clear of the patients are treatment naïve or they were included in the study after different treatments for cancer, such chemotherapy. Also, a discussion regarding the impact of chemotherapy may have on shaping bile microbiota and developing resistance to antibiotics.
[Reply]: Thank you for your suggestion. In the Introduction, we added lines 65-70, in the Discussion, the lines 283-286, and in Materials and Methods, the lines 359-362, and we added references 8 and 9.
4) It is also not clear what was the incidence of infections from hepato-pancreatic-biliary anatomic region in the patients before enrolling in the study.
[Reply]: Thank you for your suggestion. In Materials and Methods section (lines 346-348), we added
“Patients with laboratory or clinical signs of sepsis using the definitions of the Third International Consensus for Sepsis and Septic Shock (Sepsis-3) were excluded from the study [57].”
5) The authors should include in their demographic table comorbidities of patients and their corresponding treatments along with a discussion how these two parameters may influence the gut microbiota and resistance to antibiotics.
[Reply]: Thank you for your suggestion. We included in Table 1 the comorbidities. Additionally, we included the comorbidities in Methods (lines 363-366), and Discussion (lines:240-249)
6) Since gut microbiota the bile microbiota can impact each other, the study would have benefited. However, a strong discussion related to this topic should be added at least.
[Reply]: thank you for your suggestion. We added the following sentence in Discussion (lines: 240-249) and related references.
The interaction between gut microbiota and bile acids (BAs) is complex and bidirectional. BAs can influence gut microbiota composition, while gut microbiota can alter BA levels. We investigated comorbidities associated with bilio-pancreatic tumors. Mos of the comorbidities found in our patients are associated with gut dysbiosis, such as liver cirrhosis and diabetes. Gut dysbiosis can compromise the intestinal barrier, in-creasing inflammation and immune dysfunction, which may raise the risk of cancers like cholangiocarcinoma. When the barrier is disrupted and bacteria contaminate the bile ducts, toxins can promote cancer development. Bacteria such as E. coli in the biliary system and altered bile acid profiles indicate chronic inflammation and elevate the risk of bilio-pancreatic tumors.
- sun, D., Xie, C., Zhao, Y. et al. The gut microbiota-bile acid axis in cholestatic liver disease. Mol Med 30, 104 (2024). https://doi.org/10.1186/s10020-024-00830-x
- Elvevi, A., Laffusa, A., Gallo, C., Invernizzi, P., & Massironi, S. Any Role for Microbiota in Cholangiocarcinoma? A Comprehensive Review. Cells, 12(3), 370. https://doi.org/10.3390/cells12030370
7) Importantly, a discussion about the composition of gut microbiota in healthy people from the same geographic space should be added to help reader to understand the changes in gut microbiota in the study patients.
[Reply]: thank you for your suggestion. We added the following sentence in Discussion (lines:213,214) and some references were added
Recent studies have reported the gut composition in healthy Italian
- Politi, C., Mobrici, M., Parlongo, R. M., Spoto, B., Tripepi, G., Pizzini, P., Cutrupi, S., Franco, D., Tino, R., Farruggio, G., Failla, C., Marino, F., Pioggia, G., & Testa, A. Role of Gut Microbiota in Overweight Susceptibility in an Adult Population in Italy.Nutrients, 15(13), 2834. https://doi.org/10.3390/nu15132834
- Sisti, D., Pazienza, V., Piccini, F., Citterio, B., Baffone, W., Donati Zeppa, S., Biavasco, F., Prospero, E., De Luca, A., Artico, M., Taurone, S., Minelli, A., Perri, F., Binda, E., Pracella, R., Santolini, R., Amatori, S., Sestili, P., Rocchi, M. B., . . . Gobbi, P. (2022). A proposal for the reference intervals of the Italian microbiota “scaffold” in healthy adults.Scientific Reports, 12(1), 1-13. https://doi.org/10.1038/s41598-022-08000-x
- Rinninella E, Raoul P, Cintoni M, Franceschi F, Miggiano GAD, Gasbarrini A, Mele MC. What is the Healthy Gut Microbiota Composition? A Changing Ecosystem across Age, Environment, Diet, and Diseases. 2019 Jan 10;7(1):14. doi: 10.3390/microorganisms7010014. PMID: 30634578; PMCID: PMC6351938.
8) Finally, there are a lot of tinny editing issues and the manuscript needs a professional proof of English proficiency.
[Reply]: The English language has been revised by a native English speaker who was a student of a British College.
Reviewer 2 Report
Comments and Suggestions for Authors
Seasonally, bile microbiome and antibiotic resistance in patients with Hepato-Pancreatic-Biliary carcinoma: a retrospective monocentric study from 2010 to 2020
Dear authors!
Congratulations on your work. Below I will add my comments and recommendations regarding this subject!
Abstract
Please rewrite lines 43-48 of the abstract, as they are not very clear. Here’s a suggestion:
"Across all seasons, the most frequently found bacteria were Escherichia coli, Pseudomonas spp., and Enterococcus spp. Regarding antibiotic resistance, bacteria showed the highest resistance to 3GC, fluoroquinolones, aminoglycosides, fosfomycin, and piperacillin-tazobactam in the summer, and the lowest resistance in the spring, with the exception of carbapenems and colistin."
No other adjustments are necessary; it is clear enough.
Introduction
It is clear enough, but an improvement would be necessary. For example, you could add a few words about the incidence and mortality of hepatopancreatic malignancies, as well as the importance of understanding microbial carriage in order to apply specific antibiotic therapy.
Material and Method
The method of subject selection and grouping by season should be clarified. Additionally, the purpose for which the ERCP exploration was performed should be specified. Certainly, it was not performed just to collect bile and identify the implicated pathogens. The objectives of this investigation should be briefly clarified in a few words.
You could also mention how many patients underwent ERCP during this period to give an idea of the incidence and impact of infections in these patients.
Results
Tables 3 and 4 should probably be moved to the supplementary material since the important details are explained in the text.
Discussion
The discussion is clear and well-written. You have also addressed the limitations of the study.
Conclusion
It would be good if the conclusions were treated in a separate chapter to highlight the conclusions and perspectives of the study.
Author Response
REVIEWER 2
Dear authors!
Congratulations on your work. Below I will add my comments and recommendations regarding this subject!
Abstract
Please rewrite lines 43-48 of the abstract, as they are not very clear. Here’s a suggestion:
"Across all seasons, the most frequently found bacteria were Escherichia coli, Pseudomonas spp., and Enterococcus spp. Regarding antibiotic resistance, bacteria showed the highest resistance to 3GC, fluoroquinolones, aminoglycosides, fosfomycin, and piperacillin-tazobactam in the summer, and the lowest resistance in the spring, with the exception of carbapenems and colistin."
No other adjustments are necessary; it is clear enough.
[Reply]: Thank you for your suggestion. We changed the abstract.
Introduction
It is clear enough, but an improvement would be necessary. For example, you could add a few words about the incidence and mortality of hepatopancreatic malignancies, as well as the importance of understanding microbial carriage in order to apply specific antibiotic therapy.
[Reply]: Thank you for your suggestion. Sentences were added in the Introduction at lines: 52-55 and Discussion section at lines 283-289; 315-323 and some references were added.
Material and Method
The method of subject selection and grouping by season should be clarified. Additionally, the purpose for which the ERCP exploration was performed should be specified. Certainly, it was not performed just to collect bile and identify the implicated pathogens. The objectives of this investigation should be briefly clarified in a few words.
You could also mention how many patients underwent ERCP during this period to give an idea of the incidence and impact of infections in these patients.
[Reply]: Thank you for your suggestion. We changed in Methods the lines: 340-348 and 367-373,
Results
Tables 3 and 4 should probably be moved to the supplementary material since the important details are explained in the text.
[Reply]: Thank you for your suggestion. About Tables 3 and 4 the authors would prefer to leave them in the results paragraph to allow readers to quickly understand the results reported in the main text.
Discussion
The discussion is clear and well-written. You have also addressed the limitations of the study.
[Reply]: Thank you for your suggestion. We added the section
Conclusion
It would be good if the conclusions were treated in a separate chapter to highlight the conclusions and perspectives of the study.
[Reply]: Thank you for your suggestion. We added the section
Reviewer 3 Report
Comments and Suggestions for Authors
The paper focuses on the evaluation of bile microbiome and its' relationship to season change and associated antibiotic resistance. Although the data is presented in detail with regard to bacteria identified on cultures and the observed antibiotic resistance, the design of the study is flawed in my opinion. The methods section must provide more detail to better support the results of the authors, for example the study population is poorly described - if the authors included in the study only patients with hepato-billiary cancers that received ERCP due to local septic complications, the results of the bile cultures will reveal a faulty bacterial population. In most cases bile microbiota is much more varied and should not be evaluated during a septic process as it is not representative of the resident microbiota. Furthermore, bile cultures cannot reveal the entirety of the resident microbial populations, for this purpose genetic analysis is more appropriate. Another aspect that limits the purpose of the study is the collection of single samples at a single time point from different patients. For the evaluation of seasonal chamge of bile microbiota, samples should be collected at different time points from the same patient. With this in mind, considering that the authors have already collected the data of this retrospective study, the structure of the text should be adapted to the actual findings and these limitations should either be mentioned or explained as how they may not apply to the current paper.
Comments on the Quality of English LanguageThe text must be reviewed for the use of English. The text is sometimes hard to follow or unclear, revising the text would significantly increase the flow of the text and would more clearly highlight the authors ideas. For example, the title should be reformulated, as it is confusing.
Author Response
REVIEWER 3
- The paper focuses on the evaluation of bile microbiome and its' relationship to season change and associated antibiotic resistance. Although the data is presented in detail with regard to bacteria identified on cultures and the observed antibiotic resistance, the design of the study is flawed in my opinion. The methods section must provide more detail to better support the results of the authors, for example the study population is poorly described - if the authors included in the study only patients with hepato-billiary cancers that received ERCP due to local septic complications, the results of the bile cultures will reveal a faulty bacterial population.
[Reply]: Thank you for your question. We improved the whole section
- In most cases bile microbiota is much more varied and should not be evaluated during a septic process as it is not representative of the resident microbiota. Furthermore, bile cultures cannot reveal the entirety of the resident microbial populations, for this purpose genetic analysis is more appropriate.
[Reply]: Thank you for your question. The title was changed, and the sentence was added in Discussion section at lines:283-290.
- Another aspect that limits the purpose of the study is the collection of single samples at a single time point from different patients.
[Reply]: Thank you for your question.
Recently, emerging studies demonstrated that ERCP was initially intended as a diagnostic method for imaging the bile and pancreatic ducts. However, it has since evolved into a primarily therapeutic procedure, a significant development in endoscopy. This shift occurred as the risks associated with ERCP were recognized, and less invasive diagnostic alternatives, such as magnetic resonance cholangiopancreatography (MRCP) and endoscopic ultrasound (EUS), were developed.
- Sciattella P, Fornero A, Giordano SMA, De Angelis CG, Cattel F. The economic burden of post-endoscopic retrograde cholangiopancreatography (ERCP) procedure infections in Italy. Glob Reg Health Technol Assess. 2024 Dec 31;11:258-264. doi: 10.33393/grhta.2024.3186. PMID: 39822274; PMCID: PMC11736644.
- Papaefthymiou, A., Landi, R., Arvanitakis, M., Tringali, A., & Gkolfakis, P. (2025). Endoscopic retrograde cholangiopancreatography: A comprehensive review as a single diagnostic tool.Best Practice & Research Clinical Gastroenterology, 101976. https://doi.org/10.1016/j.bpg.2025.101976
- Sugimoto M, Takagi T, Suzuki T, Shimizu H, Shibukawa G, Nakajima Y, Takeda Y, Noguchi Y, Kobayashi R, Imamura H, Asama H, Konno N, Waragai Y, Akatsuka H, Suzuki R, Hikichi T, Ohira H. A new preprocedural predictive risk model for post-endoscopic retrograde cholangiopancreatography pancreatitis: The SuPER model. Elife. 2025 Jan 17;13:RP101604. doi: 10.7554/eLife.101604. PMID: 39819489; PMCID: PMC11741517.
Additionally, this aspect was reported in the Limitations section
- For the evaluation of seasonal change of bile microbiota, samples should be collected at different time points from the same patient. With this in mind, considering that the authors have already collected the data of this retrospective study, the structure of the text should be adapted to the actual findings and these limitations should either be mentioned or explained as how they may not apply to the current paper.
[Reply]: Thank you for your question. A sentence was added in the limitation section at lines 329-332
- The text must be reviewed for the use of English. The text is sometimes hard to follow or unclear, revising the text would significantly increase the flow of the text and would more clearly highlight the authors ideas. For example, the title should be reformulated, as it is confusing.
[Reply]:
The English language has been revised by a native English speaker who was a student of a British College.
Round 2
Reviewer 1 Report
Comments and Suggestions for Authors
I would like to thank the authors for their further work to respond to my comments. Nevertheless, the authors responded diligently to all comments. However, I still recommend a very careful editing. There are still tinny mistakes. Overall, in the present form, the manuscript is suitable for publication.
Author Response
REVIEWER 1
I would like to thank the authors for their further work to respond to my comments. Nevertheless, the authors responded diligently to all comments. However, I still recommend a very careful editing. There are still tinny mistakes. Overall, in the present form, the manuscript is suitable for publication.
[Reply]: Thank you for your suggestion. We performed a new editing.
Reviewer 3 Report
Comments and Suggestions for Authors
The authors have addressed most of the highlighted issues. Considering the retrospective nature of the study, the difficulty in obtaining seriate samples from the same patients and the relatively small sample size, the study highlights the need for further studies on the resident microbiota of patients with bilio-pancreatic tumors. I would recommend that the authors add the limitations regarding the methodology, the retrospective aspect.
Comments on the Quality of English LanguageEnglish was improved but is still unclear in some paragraphs. For example on page 6,lines 149-150, the authors mention that "there are more bacteria than others in every season". It is not clear who or what "others" are.
Author Response
REVIEWER 3
1) The authors have addressed most of the highlighted issues. Considering the retrospective nature of the study, the difficulty in obtaining seriate samples from the same patients and the relatively small sample size, the study highlights the need for further studies on the resident microbiota of patients with bilio-pancreatic tumors. I would recommend that the authors add the limitations regarding the methodology, the retrospective aspect.
[Reply]: Thank you for your suggestion. The methodology used in the Limitations section has already been reported (lines 340-348). Additionally, we added/improved (lines: 349-354), adding the retrospective aspect of the study and the impact due to a small sample on some no significant results obtained.
2) English was improved, but some paragraphs are still unclear. For example, on page 6, lines 149-150, the authors mention that "there are more bacteria than others in every season". It is not clear who or what "others" are.
[Reply]: Thank you for your suggestion. We changed the sentence and performed further English language checking.